# Olfactory Dysfunction Is Associated with Cerebral Amyloid Deposition and Cognitive Function in the Trajectory of Alzheimer’s Disease

**DOI:** 10.3390/biom13091336

**Published:** 2023-08-31

**Authors:** Sheng-Min Wang, Dong Woo Kang, Yoo Hyun Um, Sunghwan Kim, Chang Uk Lee, Hyun Kook Lim

**Affiliations:** 1Department of Psychiatry, Yeouido St. Mary’s Hospital, College of Medicine, The Catholic University of Korea, Seoul 06591, Republic of Korea; 2Department of Psychiatry, Seoul St. Mary’s Hospital, College of Medicine, The Catholic University of Korea, Seoul 06591, Republic of Korea; 3Department of Psychiatry, St. Vincent Hospital, Suwon, Korea, College of Medicine, The Catholic University of Korea, Suwon 16247, Republic of Korea

**Keywords:** olfaction, Alzheimer’s disease, mild cognitive impairment, beta-amyloid, association

## Abstract

Olfactory dysfunction is consistently observed in individuals with Alzheimer’s disease (AD), but its association with beta-amyloid (Aβ) deposition remains unclear. This study aimed to investigate the relationship among olfactory function, cerebral Aβ deposition, and neuropsychological profiles in individuals with no cognitive impairment (NCI), mild cognitive impairment (MCI), and AD dementia. A total of 164 participants were included, and olfactory function was assessed using the brief smell identification test (B-SIT). Cerebral Aβ deposition was measured using [^18^F]-flutemetamol PET imaging (A-PET). The results show a significant group difference in olfactory function, with the highest impairment observed in the Aβ-positive MCI and AD dementia groups, and the impairment was the lowest in Aβ-negative NCI. Olfactory dysfunction was positively associated with cognitive impairments across multiple domains. Furthermore, individuals with Aβ deposition had lower olfactory function compared to those without Aβ, even within the same neuropsychological stage. The association between olfactory dysfunction and Aβ deposition was observed globally and in specific cortical regions. These findings suggest that olfactory dysfunction is associated with both cognitive function and cerebral Aβ pathology in the trajectory of AD. Olfactory deficits may serve as an additional marker for disease progression and contribute to understanding the underlying mechanisms of AD.

## 1. Introduction

Alzheimer’s disease (AD) is a progressive neurodegenerative disorder characterized by a gradual and long-term decline in cognitive function [1]. A contemporary amyloid cascade hypothesis suggests that deposition of beta-amyloid (Aβ) peptide is an upstream event that is associated with downstream tau deposition, neurodegeneration, and eventual cognitive decline [2]. Clinico-pathophysiological changes associated with AD begin decades before clinical symptom onset [3]. Individuals with mild cognitive impairment (MCI), representing a transitional stage between normal aging and AD, have objective cognitive decline but intact activities of daily living, and a substantial proportion of individuals with MCI eventually progress to dementia [4]. With the recent development of a disease-modifying drug for AD, and more potential drugs targeting AD at its earlier phase before the clinical symptoms become evident, determining the individuals who are at risk of AD and predicting their disease course became even more important [5,6]. There was significant progress in understanding the pathophysiology of AD, including the roles of Aβ, tau, and neuroinflammation [7]. However, the discovering of novel biomarkers for AD remains important because these known biomarkers cannot completely detect onset, predict progression, or monitor symptoms of AD [8,9].

Olfactory deficits, or lower ability to identify odor, are consistently observed in individuals with the trajectory of AD [10]. A study composed of multi-ethnic older adults showed that individuals with MCI have poorer odor identification ability than individuals with normal cognition or cognitive normal older adults (CN) [11]. Others showed that olfactory deficit was more pronounced in patients with probably AD, based on the National Institute of Neurological and Communicative Disorders and Stroke–Alzheimer’s Disease and Related Disorders Association (NINCDS-ADRDA) criteria, than in patients with MCI or CN [12]. Such findings were repeated in meta-analysis, which found that compared with CN, patients with MCI [13] or probably AD [14] have lower ability to identify odor. Longitudinal studies further showed that CN with olfactory impairments were more likely to develop MCI than those having intact odor identification ability [15,16]. Likewise, patients with MCI having an olfactory deficit had a higher risk of exhibiting progressive cognitive decline and converting to dementia due to AD or dementia with cerebral Aβ deposition (AD dementia) [17].

Multiple studies indicated that there is a neurobiological link between olfactory dysfunction and AD pathology. Post-mortem studies indicated that the pathological changes in AD occur in early stage of disease in entorhinal and transentorhinal areas, which overlap with the brain regions involved in olfactory processing [18,19]. Moreover, reduced volume of the hippocampus and medial temporal lobe were found to be associated with olfactory dysfunction in CN, MCI, and AD dementia [20,21]. Tau pathology is known to precede neurodegeneration, and a study showed cerebrospinal fluid level of total-tau and P_181_-tau were associated with olfactory deficit in CN who are a at increased risk of AD [22]. More recent research indicated that the atrophy and the decreased volumes of brain regions affected by tau pathology correlated with olfactory impairment [23].

Despite the above findings, which showed a close link between the olfactory dysfunction and various AD pathologies, the association between olfactory deficit and cerebral Aβ deposition, one of the most important biomarkers reflecting the earliest disease stage of AD, is still not clear [24]. A study using ^11^C-labeled Pittsburgh Compound B demonstrated a positive correlation between olfactory dysfunction and cerebral Aβ deposition in CN [20]. In another study, however, MCI patients with high Aβ deposition and MCI patients with low Aβ deposition did not differ in their odor identification abilities [25]. A more recent study showed no difference in the Aβ burden between the normosmia and hyposmia groups and no correlation between severity of olfactory dysfunction with Aβ burden [26]. A meta-analysis also showed that olfactory dysfunction was associated Aβ deposition in the CN group but not in the MCI group [27].

A possible explanation for these contradictory results could be attributed to diverse co-factors, which possibly influenced the relationship between olfactory function and Aβ deposition. In one study, the sample size was not sufficient to investigate within group difference of olfactory function according to the Aβ deposition (for example, n = 14 for Aβ positive MCI and n = 10 for Aβ negative MCI) [25]. Age is one of the most important factors influencing olfactory function independent of AD pathology [28], but previous studies did not take this into account and had significant group difference in age [25,26]. A relatively smaller proportion of subjects having high Aβ burden relative to low Aβ burden was another important drawback. Thus, studies failed to elucidate whether the olfactory dysfunction is due to age difference or Aβ pathology per se.

To fill in this gap, the present study aimed to investigate the relationship between olfactory function, cerebral Aβ deposition, and neuropsychological profiles in individuals with no cognitive impairment (NCI), MCI, and AD dementia. We hypothesized that, having a similar age range and rate of subjects with high Aβ deposition among groups, the olfactory dysfunction would show an association with both cerebral Aβ deposition and neuropsychological profiles.

## 2. Materials and Methods

### 2.1. Subjects

A total of 164 subjects, consisting of 44 NCI individuals with negative results in amyloid position emission tomography (A-PET(−)), 30 A-PET positive (A-PET (+)) NCI, 34 A-PET (−) MCI patients, 31 A-PET (+) MCI patients, and 25 A-PET (+) dementia patients (AD dementia), were included in the study. Subjects were recruited from volunteers in the Catholic Aging Brain Imaging (CABI) database, which contains the brain scans of patients who visited the outpatient clinic at Catholic Brain Health Center, Yeouido St. Mary’s Hospital, The Catholic University of Korea, between 2017 and 2022. The inclusion criteria applied commonly for all subjects were as follows: (1) they must be age ≥55 years and (2) have no clinically significant psychiatric disorders (depressive disorder, schizophrenia, or bipolar disorder).

In terms of NCI groups, they visited our outpatient clinic with subjective complaints of cognition. Their normal cognitive functions were confirmed with the Korean version of the Consortium to Establish a Registry for Alzheimer’s Disease (CERAD-K), which includes a verbal fluency (VF) test, the 15-item Boston Naming Test (BNT), the Korean version of the Mini-Mental State Examination (MMSE), word list memory (WLM), word list recall (WLR), word list recognition (WLRc), constructional praxis (CP), and constructional recall (CR) [29]. The patients with MCI groups met the following criteria: (1) memory complaints corroborated by an informant; (2) at least 1.0 standard deviation (SD) below age- and education-adjusted norms in more than one cognitive domain on the CERAD-K, (3) intact activities of daily living; (4) global clinical dementia rating score (CDR) of 0.5; and (5) not demented according to the Diagnostic and Statistical Manual of Mental Disorders (DSM)-V criteria. Patients in the AD dementia group (1) had global CDR score of >1 and (2) met the probable AD criteria proposed by the NINCDS- ADRDA [30] (3) as well as those proposed by the Diagnostic and Statistical Manual of Mental Disorders, fifth edition, with A-PET positive results [31].

### 2.2. Assessment of Odor Identification

All subjects received an olfactory function test or odor identification ability assessment using the 12-item brief smell identification test (B-SIT). The B-SIT, which is an abridged test of olfaction derived from the 40-item University of Pennsylvania Smell Identification Test, was shown to be valid in cross-cultural settings [32]. The detailed procedures are described elsewhere [33]. To be brief, the B-SIT is a booklet for a standardized, 12-item, 4-alternative forced-choice measure. Each page contains a scratchable patch of microencapsulated odorant. To conduct the test, the examiner used a pencil to scratch the odor patch, releasing the odorant. Thereafter, the patch was placed beneath the participant’s nose, and they were asked to identify the specific odor that closely resembled the item. The participant’s score was determined by the number of correctly identified odors, which ranged from 0 to 12. In cases where responses to one or two items were missing, each missing response was assigned a score of 0.25. If three or more items had missing responses, the data for the test were considered incomplete.

### 2.3. [^18^F]-Flutemetamol PET Image Acquisition and Processing

We followed previous studies for data collection and analytic procedures for [^18^F]-flutemetamol (^18^F-FMM) and ^18^F-FMM PET [34]. The analysis of ^18^F-FMM PET data was based on the standardized uptake value ratio (SUVR) 90 min post injection. We measured six cortical regions of interest (frontal, superior parietal, lateral temporal, striatum, anterior cingulate cortex, and posterior cingulate cortex/precuneus) using the PMOD Neuro Tool to extract regional SUVR values. Thereafter, we averaged the SUVR values of these six cortical regions of interest to calculate the global Aβ burden or global SUVR values. In accordance with previous studies, we used a neocortical SUVR of 0.62 as the cutoff between high and low Aβ burden [34]. However, amyloid positivity was confirmed by visual reading from two separate nuclear medicine radiologists.

### 2.4. Statistical Analysis

A free and open-source data analysis tool, Jamovi (Version 2.3.18.0), was utilized to perform statistical analyses [35]. Analysis of variance (ANOVA) and the chi-square test were used to assess statistical differences among five groups for continuous variables and the categorical variables, respectively. When the group difference was statistically significant, we used Bonferroni tests for post-hoc analysis and multiple corrections. In terms of association studies, Pearson correlation analysis was utilized to investigate association among continuous variables. In all analyses, a two-tailed α level of 0.05 was chosen to indicate statistical significance.

## 3. Results

### 3.1. Baseline Demographic and Clinical Data

The baseline demographic data of the study (*n* = 164) are presented in Table 1. All variables were normally distributed, and there were no significant differences in age, education, and sex among all groups. In terms of NCI group, the A-PET (+) NCI group had a higher rate of ε4 and cerebral Aβ deposition, the mean global SUVR values, than the A-PET (−) NCI group, but the two groups did not show statistical difference in neuropsychological profiles. Likewise, A-PET (+) MCI and A-PET (−) MCI groups did not show statistical difference in neuropsychological profiles, but the A-PET (+) MCI group had higher-rate of ε4 and global SUVR values than the A-PET (−) MCI group. The AD dementia group had the lowest scores of neuropsychological profiles. The SUVR values between A-PET (−) NCI and A-PET (−) MCI did not differ. Lastly, the AD dementia had numerically the highest global SUVR value, but the global SUVR values did not differ among A-PET (+) NCI, A-PET (+) MCI, and AD dementia groups.

### 3.2. Group Difference in Olfactory Function

There was a group difference in olfactory function or mean B-SIT scores (*p* < 0.001 for ANOVA). Post-hoc analysis showed that the B-SIT score was highest in the A-PET (−) NCI and was the lowest in the A-PET (+) MCI and AD dementia groups (for both *p* < 0.05 Bonferroni corrected). In addition, among the patients having the same neuropsychological staging, B-SIT score was also significantly higher in the A-PET (−) groups than in the A-PET (+) groups (A-PET (−) NCI > A-PET (+) NCI and A-PET (−) MCI > A-PET (+) MCI, for all *p* < 0.05 Bonferroni corrected). However, there were no differences in B-SIT scores between the A-PET (+) MCI and AD dementia groups (Figure 1 for all group difference results).

### 3.3. Association between Olfactory Function with Cerebral Aβ Deposition and Neuropsychological Profiles

Correlation analysis between olfactory function and neuropsychological profiles showed that the B-SIT had a positive association with all the sub-scores of and total score of CERAD-K (Figure 2A–I).

Among 164 subjects, the total number of subjects with A-PET (+) and A-PET (−) were 86 and 78, respectively. Before conducting association analysis between the olfactory function and Aβ deposition, we re-grouped the participants into A-PET (+) and A-PET (−) groups. The two groups did not differ in age (75.52 ± 6.98 vs. 74.54 ± 5.74, p = 0.328), but the A-PET (+) group had statistically lower B-SIT scores than the A-PET (−) group (6.19 ± 2.82 vs. 8.09 ± 2.17, *p* < 0.001) (Figure 3).

Figure 4 shows the results of correlation analysis between the B-SIT scores and the cerebral Aβ deposition level (SUVR). Global SUVR showed a negative association with the B-SIT scores (r = −0.305, *p* < 0.001). Among the six cortical regions, regional SVURs of the PCC/PC (r = −0.312, *p* < 0.001), frontal lobe (r = −0.270, *p* < 0.001), parietal lobe (r = −0.267, *p* < 0.001), and lateral temporal lobe (r = −0.203, *p* < 0.01) showed a negative association with the B-SIT scores (Figure 4A–E).

We also conducted linear regression analysis to investigate the associations of baseline demographic and clinical factors including age, sex, education, total score of the CERAD-K, clinical diagnosis, and global SUVR scores with the B-SIT scores. The results show that the younger age, female, higher total score of the CERAD-K, and lower global SUVR scores were associated with higher B-SIT score (R = 0.585, *p* < 0.01) (Table 2).

## 4. Discussion

The present study aimed to investigate the relationship between olfactory function, cerebral Aβ deposition, and neuropsychological profiles in individuals with NCI, MCI, and AD dementia. Consistent with previous research, our results demonstrate a significant group difference in olfactory function across all stages of cognitive impairment [13,14]. Olfactory deficits, as measured by the B-SIT scores, were most severe in the A-PET (+) MCI and AD dementia groups, while the A-PET (−) NCI group exhibited the highest olfactory function. These findings align with the notion that olfactory impairment worsens as individuals progress from normal cognition to dementia [10].

We also observed significant associations between olfactory function and neuropsychological profiles. Consistent with previous studies, the B-SIT scores showed positive associations the CERAD-K scores, which suggests that individuals with poorer olfactory function are likely to exhibit more severe cognitive impairments across multiple domains [36]. The olfactory function could be decreased simply due to a decline in (odor) memory domain [37]. However, we showed that the B-SIT scores showed positive associations with not only the total score, but also with all the sub-scores of the CERAD-K, including memory and non-memory domains. Thus, our research suggests that the odor identification requires multiple subsets of cognitive functions, including attention, naming, execution, and memory [10].

In the other perspective, olfactory dysfunction might be due to cognitive decline only rather than due to the AD pathology. In line with this theory, olfactory dysfunction increases with age and is highly prevalent in those with diverse neurodegenerative conditions, including head trauma [38], Parkinson’s disease [39], dementia with Lewy bodies [40], and frontotemporal dementia [41]. Studies also suggested that the olfactory dysfunction is associated with tau-pathology, which is increased not only in AD, but also in diverse degenerative disorders affecting the brain, and neurodegeneration [18,20,21,22]. However, conflicting results were reported regarding its correlation with cerebral Aβ [20,25,26], which is a hallmark of the AD [24].

Taking these together, we explored olfactory function differences according to the cerebral Aβ deposition severity. Previous results show that A-PET (+) MCI and A-PET (−) MCI did not differ in olfactory identification ability [20,42]. In contrast, we observed that within same neuropsychological stage, the patients with Aβ (+) groups had lower olfactory function than the patients with Aβ (−) groups (A-PET (+) NCI < A-PET (−) NCI; A-PET (+) MCI < A-PET (−) MCI). The previous studies had an unbalanced mean age between A-PET (+) and A-PET (−) groups [25,26], whereas all five groups in our study had comparable age (mean 75.03 ± 6.39, range 74.41 ± 5.11~75.40 ± 7.95). By excluding age, which is an important factor influencing olfactory function, as a possible cofactor [28], we were able to investigate olfactory change associated with the Aβ pathology more clearly. In this perspective, our results of total A-PET (+) and A-PET (−) groups having similar age (75.52 ± 6.98 vs. 74.54 ± 5.74, *p* = 0.328) but showing statistical difference in B-SIT scores (6.19 ± 2.82 vs. 8.09 ± 2.17, *p* < 0.001) are also noteworthy.

Our results confirm those of previous studies that found a positive association between olfactory function and cerebral Aβ deposition [17]. Since the earlier study included patients with NCI only, the study was not able to elucidate whether the association persisted throughout the trajectory of AD or was found only in the preclinical stage of AD. Thus, we advanced previous research by showing association between Aβ deposition and olfactory dysfunction in patients along the AD continuum, ranging from A-PET (−) NCI to AD dementia, with cerebral Aβ status defined using A-PET and neurocognitive function measured using CERAD-K. Moreover, we provided the novel finding that the olfactory dysfunction had positive association not only with the global Aβ deposition, but also with the regional Aβ deposition of four cortical areas, including PC/PCC, the frontal lobe, parietal lobe, and temporal lobe. These results suggest that the presence of both global and regional cerebral Aβ pathology might contribute to olfactory dysfunction in the trajectory of AD. Linear regression analysis showed that the younger age, being female, higher total score of the CERAD-K, and lower global SUVR scores were associated with higher B-SIT scores. The results might suggest that the AD pathology or cerebral Aβ deposition has a detrimental effect on the olfactory function in addition to age, sex, and cognitive functions. Thus, within the same neurocognitive staging, the Aβ deposition might contribute to additional olfactory dysfunction. In the other perspective, among individuals having high or positive Aβ deposition, the cognitive function might impact further olfactory dysfunction. Considering these factors collectively, after an individual received baseline evaluations of AD-related biomarkers (i.e., A-PET) and neuropsychological profiles, the longitudinal analysis of B-SIT might be a quick and an economic method to reflect cognitive decline over time in individuals with NCI or MCI. Similarly, lower B-SIT scores relative to the individual’s cognitive function might suggest presence of Aβ deposition. Nevertheless, additional studies are required to confirm our speculations.

It is not clear why the olfactory dysfunction did not show association with the regional Aβ deposition of the striatum and the anterior cingulate cortex. The olfactory system has direct connections with specific brain regions involved in olfactory processing, such as the olfactory bulb followed by the prepiriform cortex, amygdala, entorhinal cortex, and hippocampus [10]. On the other hand, the striatum and the anterior cingulate cortex have different functional, such as emotional processing, and have a limited role in olfactory processing [43]. Thus, the regional Aβ deposition in these areas may not have directly impact the olfactory pathway, which could explain the lack of association with olfactory dysfunction. However, further studies investigating association among functional and structural connectivity of the olfactory pathway, the regional Aβ deposition, and olfactory function are needed to clarify this issue.

Our study has additional strengths. By including a similar proportion of subjects having both low (*n* = 78) and high (*n* = 86) cerebral Aβ deposition, we were able to investigate olfactory dysfunction associated with Aβ more clearly. The careful selection criteria enabled us to include older adults (average age higher than 75) with balanced baseline demographic data among five groups within the trajectory of AD. Thus, we were able to prevent diverse confounding factors. In addition, we used a single tracer, ^18^F-FMM, to investigate the cerebral Aβ deposition and prevented possible bias arising from the inter-tracer variability.

This study also has several limitations. We used a cross-sectional design, so we were able to investigate associations only and have limited ability to infer causal pathways among olfactory dysfunction, Aβ pathology, and cognitive decline. Second, all subjects were from a single center, which limits the generalizability of our results. Third, patients having dementia due to non-AD pathologies were also known to show olfactory dysfunction, but we did not include patients with A-PET (−) dementia. Thus, we were not able to investigate whether olfactory dysfunction differs in patients with dementia depending on the Aβ pathology. In addition, we showed no differences in B-SIT scores between the A-PET (+) MCI and AD dementia groups. Our results might indicate that the patients with late-MCI to moderate dementia could show similar olfactory function because of their cognitive impairments. In this perspective, the clinical utility of B-SIT may not be high in patients with more severe cognitive dysfunction. Additional studies are needed to overcome this limitation. We were unable to investigate tau pathologies using either tau-PET or CSF studies or neurodegeneration using T1 MRI, which means that the difference of olfactory function we observed could be influenced by tau pathologies as well as neurodegeneration. Thus, further longitudinal studies are needed to elucidate the causal or sequential relation among olfactory dysfunction, Aβ and tau pathology, neurodegeneration, and cognitive decline in the trajectory of AD.

## 5. Conclusions

Our study demonstrated a significant group difference in olfactory function across all stages of cognitive impairment. We also showed that the olfactory dysfunction was associated with multiple subsets of and total scores of cognitive functions. We explored olfactory function differences according to the cerebral Aβ deposition severity and showed that within the same neuropsychological stage, the patients with Aβ (+) groups had lower olfactory function than the patients with Aβ (−) groups. Finally, we showed that the olfactory dysfunction was associated with cerebral Aβ deposition in individuals with NCI, MCI, and AD dementia. Olfactory deficits may serve as an additional marker for cerebral Aβ deposition and cognitive function, especially in non-demented individuals. Further research is warranted to elucidate the underlying mechanisms and clinical implications of olfactory dysfunction in AD, with the ultimate goal of improving early detection and management of the disease.

## Figures and Tables

**Figure 1 biomolecules-13-01336-f001:**
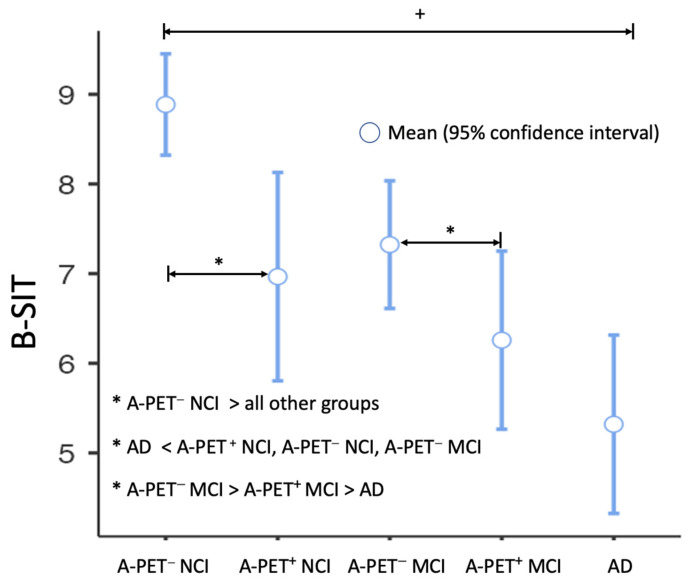
Group difference in B-SIT. + *p* < 0.001 in ANOVA, * *p* < 0.05 for post-hoc analysis with Bonferroni correction. A-PET: Amyloid-positron emission tomography using 18F-flutemetamol; AD: Alzheimer’s disease; B-SIT: brief smell identification test; NCI: no cognitive impairment; and MCI: mild cognitive impairment.

**Figure 2 biomolecules-13-01336-f002:**
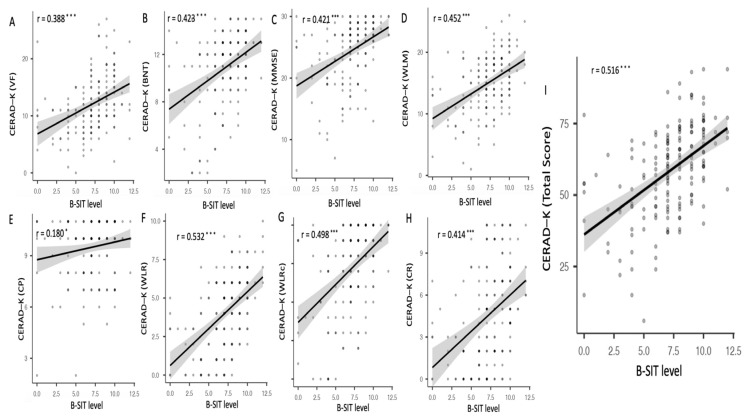
Association between B-SIT level and neuropsychological measures; * *p* < 0.05, *** *p* < 0.001; B-SIT: brief smell identification test; CERAD-K: The Korean Version of Consortium to Establish A Registry for Alzheimer’s Disease. There was positive association between scores of CERAD-K and B-SIT scores for (**A**) VF: verbal fluency (**B**) BNT: Boston naming test (**C**) MMSE: mini mental status examination, (**D**) WLM: word list memory, and (**E**) CP: constructional praxis, (**F**) WLR, word list recall, (**G**) WLRc: word list recognition, (**H**) CR: constructional recall, (**I**) CERAD-K total score.

**Figure 3 biomolecules-13-01336-f003:**
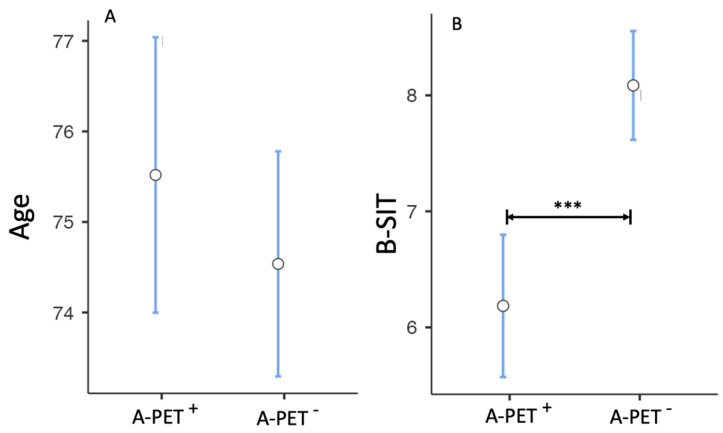
Age and B-SIT score differences between A-PET (+) and A-PET (−) groups; *** *p* < 0.001; (**A**) the two groups did not show difference in age, but (**B**) B-SIT scores were statically higher in the A-PET (+) group than in the A-PET (−) group.

**Figure 4 biomolecules-13-01336-f004:**
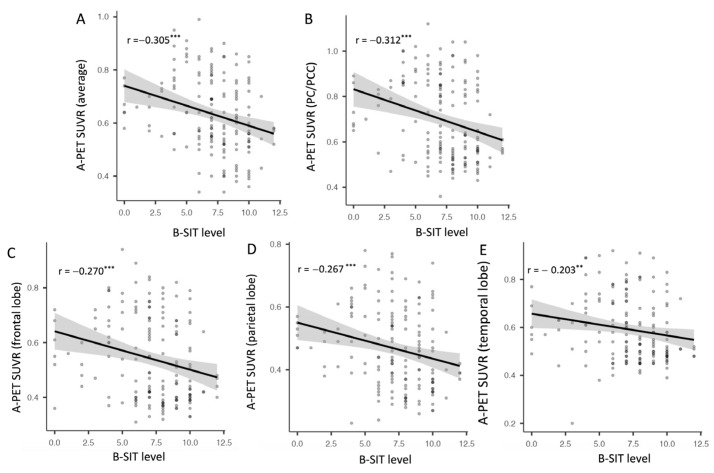
Association between B-SIT level and global and regional cerebral Aβ deposition; ** *p* < 0.01, *** *p* < 0.001; B-SIT: brief smell identification test; PC: precuneus; PCC: posterior cingulate cortex; and SUVR: standardized uptake volume ratio. There was negative association between SUVR scores and B-SIT scores for (**A**) global cerebral Aβ deposition and regional Aβ deposition in (**B**) PC/PCC, (**C**) the frontal lobe, (**D**) parietal lobe, and (**E**) temporal lobe.

**Table 1 biomolecules-13-01336-t001:** Demographic and clinical characteristics of the study participants.

	Total (*n* = 164)	A-PET (−) NCI (*n* = 44)	A-PET (+) NCI (*n* = 30)	A-PET (−) MCI (*n* = 34)	A-PET (+) MCI (*n* = 31)	AD Dementia (*n* = 25)	*p*-Value
Age (years ± SD)	75.03 (6.39)	74.41 (5.11)	74.20 (6.90)	75.32 (6.17)	75.85 (6.95)	75.40 (7.95)	NS
Education (years ± SD)	11.49 (4.30)	11.80 (4.34)	12.08 (3.67)	10.67 (4.13)	11.35 (5.21)	11.00 (4.00)	NS
Sex (M:F)	50:114	12:32	10:20	10:24	10:21	8:17	NS
SUVR	0.63 (0.154)	0.50 (0.074)	0.71 (0.08)	0.51 (0.068)	0.74 (0.087)	0.78 (0.11)	<0.001
APOE4 N(%)	52 (31.7%)	10 (22.7%)	18 (60%)	8 (23.5%)	16 (51.6)	10 (40%)	<0.001
CDR		0	0	0.5	0.5	1	
CERAD-K Battery (SD)							
VF	12.06 (5.1)	14.73 (4.09)	14.96 (4.88)	10.79 (4.13)	11.68 (3.45)	6.40 (3.25)	<0.001
BNT	10.75 (3.21)	12.59 (1.77)	12.26 (1.60)	9.59 (2.99)	10.13 (3.06)	8.08 (3.85)	<0.001
MMSE	24.41 (5.12)	28.17 (1.69)	27.84 (1.60)	24.12 (3.51)	23.25 (3.47)	16.35 (5.16)	<0.001
WLM	14.95 (4.65)	18.39 (3.18)	17.26 (3.24)	14.03 (3.45)	13.84 (3.25)	8.84 (4.17)	<0.001
CP	9.44 (1.78)	10.07 (1.19)	10.20 (0.89)	8.85 (1.72)	9.55 (1.67)	8.10 (2.60)	<0.001
WLR	4.10 (2.42)	6.21 (1.45)	5.53 (1.52)	3.48 (1.84)	2.894 (1.79)	1.00 (1.45)	<0.001
WLRc	7.29 (2.73)	9.15 (1.03)	8.68 (1.31)	7.44 (2.21)	6.42 (2.51)	3.32 (2.41)	<0.001
CR	4.54 (3.44)	6.52 (2.78)	6.53 (3.11)	4.06 (2.81)	3.10 (3.30)	1.16 (2.06)	<0.001
Totalscore	58.61 (16.24)	71.09 (9.60)	70.1 (10.22)	54.27 (11.2)	54.76 (10.27)	34.95 (13.9)	<0.001
B-SIT (SD)	7.16 (2.67)	8.89 (1.86)	6.97 (3.11)	7.32 (2.04)	6.26 (2.71)	5.32 (2.41)	<0.001

A-PET: amyloid positron emission tomography using ^18^F-flutemetamol; AD: dementia due to Alzheimer’s disease; APOE: apolipoprotein E; BNT: 15-item Boston Naming Test; B-SIT: 12-item brief smell identification test; CERAD-K: The Korean Version of Consortium to Establish A Registry For Alzheimer’s Disease; CDR: clinical dementia rating; CP: constructional praxis; CR: constructional recall; MCI: mild cognitive impairment; MMSE: Mini Mental Status Examination; NCI: no cognitive impairment; NS: not significant, SD: standard deviation; SUVR: standardized uptake value ratio with ^18^F-flutemetamol; VF: verbal fluency; WLRc: word list recognition; WLM: word list memory; and WLR, word list recall.

**Table 2 biomolecules-13-01336-t002:** Results of multiple linear regression analysis investigating the associations of age, sex, education, total scores of the CERAD-K, clinical diagnosis, and global SUVR scores with B-SIT scores.

Independent Variable	Estimate	SE	*t*-Statistic	*p*-Value
Intercept	9.6970	3.0932	3.135	0.002
Age	−0.0776	0.0301	−2.578	0.011
Female sex	0.8301	0.4176	1.988	0.049
Education	−0.0626	0.0509	−1.231	0.220
CERAD-K total score	0.0849	0.0195	4.359	<0.001
Clinical diagnosis				
MCI	0.2979	0.4968	0.600	0.550
AD dementia	1.2587	0.9290	1.355	0.177
Global SUVR	−2.9035	1.3823	−2.100	0.037

B-SIT: 12-item brief smell identification test; CERAD-K: The Korean Version of Consortium to Establish A Registry for Alzheimer’s Disease; MCI: mild cognitive impairment; and SUVR: standardized uptake volume ratio.

## Data Availability

The datasets generated for this study are available on request to the corresponding author.

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
