# Peer review of "Olfactory Dysfunction Is Associated with Cerebral Amyloid Deposition and Cognitive Function in the Trajectory of Alzheimer’s Disease"

_biomolecules, 2023, doi:10.3390/biom13091336_

Round 1

Author Response

Author response)

 Thank you very much for your comprehensive review of our paper. We would also like to thank you for your encouraging comments. In order to make our paper more logical and valid, we made corrections based your comments. The responses to your comments are appended by point-to-point and all the changes are indicated in the revised version with yellow highlights.

Reviewer 2 Report

This paper was well and scientifically written. However, I found a few points that need to be considered to make this paper better. 

1. In this paper, there were no differences in B-SIT scores between the A-PET (+) MCI and AD dementia groups. Therefore, the title "Olfactory Dysfunction is Associated with ....and Disease Severity in Alzheimer’s Disease" is incorrect and need to be modified. In conclusions, "Olfactory deficits appear to parallel the progression of AD pathology and may serve as an additional marker for disease progression" should also be corrected. 

2. Please show the B-SIT scores of A-PET (-) dementia and compare with other groups. 

3. Olfactory test may be impossible for moderate to sever dementia patients because of their cognitive impairments.  And in your study,  there were no differences in B-SIT scores between the A-PET (+) MCI and AD dementia groups. Therefore olfactory test may be useful in MCI stage. Please describe this limitation in the discussion. 

4.  Please show the results of the comparisons between groups in table 1. 

Spelling errors in lines 143, 197, and 218. 

Author Response

(The authors gave the same response as above.)

Round 2

Reviewer 2 Report

None